# Emission of coherent THz magnons in an antiferromagnetic insulator triggered by ultrafast spin–phonon interactions

E. Rongione [1,2], O. Gueckstock [3], M. Mattern[4], O. Gomonay [5], H. Meer[5], C. Schmitt[5], R. Ramos[6,7], T. Kikkawa [8], M. Mičica[2], E. Saitoh[6,8,9], J. Sinova[5], H. Jaffrès [1], J. Mangeney[2], S. T. B. Goennenwein [10], S. Geprägs[11], T. Kampfrath[3], M. Kläui [5,12,13], M. Bargheer [4,14], T. S. Seifert[3] ✉, S. Dhillon[2] & R. Lebrun [1] ✉

Antiferromagnetic materials have been proposed as new types of narrowband THz spintronic devices owing to their ultrafast spin dynamics. Manipulating coherently their spin dynamics, however, remains a key challenge that is envisioned to be accomplished by spin-orbit torques or direct optical excitations. Here, we demonstrate the combined generation of broadband THz (incoherent) magnons and narrowband (coherent) magnons at 1 THz in low damping thin films of NiO/Pt. We evidence, experimentally and through modeling, two excitation processes of spin dynamics in NiO: an off-resonant instantaneous optical spin torque in (111) oriented films and a strain-wave-induced THz torque induced by ultrafast Pt excitation in (001) oriented films. Both phenomena lead to the emission of a THz signal through the inverse spin Hall effect in the adjacent heavy metal layer. We unravel the characteristic timescales of the two excitation processes found to be < 50 fs and > 300 fs, respectively, and thus open new routes towards the development of fast opto-spintronic devices based on antiferromagnetic materials.

Antiferromagnetic spintronics has recently become an important research field from both a fundamental viewpoint and its strong applicative potential[1,2]. Antiferromagnets (AFM) have key advantages linked to their magnetic ordering: they are insensitive to perturbative external magnetic fields, stray fields are absent, and magnon modal frequencies reach the terahertz (THz) regime[1–3]. This renders AFMs

prime candidates for ultrafast spintronic devices[4] compared to their ferromagnetic counterparts. A recent work demonstrated the writing of AFM memory states with picosecond excitations[5]. In parallel, narrowband sub-THz detection has been achieved using spin-pumping in AFMs[6,7]. Another spintronic application is spintronic-based broadband THz emission that currently relies on ferromagnet/heavy metal

[1]Unité Mixte de Physique, CNRS, Thales, Université Paris-Saclay, F-91767 Palaiseau, France. [2]Laboratoire de Physique de l'Ecole Normale Supérieure, ENS, Université PSL, CNRS, Sorbonne Université, Université Paris Cité, F-75005 Paris, France. [3]Institute of Physics, Freie Universität Berlin, D-14195 Berlin, Germany. [4]Institut für Physik und Astronomie, Universität Potsdam, D-14476 Potsdam, Germany. [5]Institute of Physics, Johannes Gutenberg-University Mainz, D-55099 Mainz, Germany. [6]WPI-Advanced Institute for Materials Research, Tohoku University, Sendai J-980-8577, Japan. [7]Centro Singular de Investigación en Química Bilóxica e Materiais Moleculares (CIQUS), Departamento de Química-Física, Universidade de Santiago de Compostela, Santiago de Compostela 15782, Spain. [8]Department of Applied Physics, The University of Tokyo, Tokyo J-113-8656, Japan. [9]Institute for AI and Beyond, The University of Tokyo, Tokyo J-113-8656, Japan. [10]Department of Physics, University of Konstanz, D-78457 Konstanz, Germany. [11]Walther-Meißner-Institut, Bayerische Akademie der Wissenschaften, D-85748 Garching, Germany. [12]Graduate School of Excellence Materials Science in Mainz (MAINZ), Staudingerweg 9, D-55128 Mainz, Germany. [13]Center for Quantum Spintronics, Department of Physics, Norwegian University of Science and Technology, N-7034 Trondheim, Norway. [14]Helmholtz-Zentrum Berlin für Materialien und Energie, Wilhelm-Conrad-Röntgen Campus, BESSY II, Albert-Einstein-Strasse 15, D-12489 Berlin, Germany. ✉e-mail: tom.seifert@fu-berlin.de; romain.lebrun@cnrs-thales.fr

heterostructures[8-13] and harnesses spin-to-charge-current conversion through the inverse spin Hall effect (ISHE). In this regard, AFM materials with their characteristic THz resonant modes have been predicted to enable the development of narrowband THz spintronic emitters, i.e., THz nano-oscillators, which can be driven by spin-orbit torques[14]. A recent work observed broadband THz emission in AFM thin films, triggered by an off-resonant optical torque[15]. Therefore, the targeted narrowband emission of coherent THz magnons, which is highly desirable to fully functionalize spintronic THz emission, remains to be shown.

Electrical switching of antiferromagnets recently highlighted that spin–orbit torques often compete with dominant thermo-magneto-elastic processes[16] and that thermally driven strain gradients can be used to reorient the antiferromagnetic domains[16]. Dynamic strain could thus be used to control antiferromagnetic states and potentially the AFM dynamics on ultrafast timescales[17-19]. Using this approach, one can envision to generate THz magnons using ultrafast strain gradients to achieve a time-dependent modulation of the exchange interaction.

In this paper, we report the combined generation of narrowband coherent THz emission centered at 1 THz and incoherent broadband THz magnons in NiO/Pt bilayers. For this purpose, we use femtosecond near-infrared laser pulses to either trigger direct and off-resonant light-spin interactions in (111) oriented films[20,21], or phonon–spin interactions in (001) oriented films through an ultrafast strain pulse that is generated upon heating of the metallic layer[22-24]. We demonstrate that their respective efficiencies in generating THz spin-currents depend on the orientation and thickness of the AFM films. We identify the processes via the anisotropic or isotropic dependence on the pump polarization and show that the THz emission process arises in both cases from ultrafast spin-to-charge-current conversion in the heavy metal layer[15,25]. Finally, we quantify the spin-current rise and decay times to be less than 50 fs, limited by the experimental resolution, for the direct light-spin coupling pathway, and to be more than 300 fs for the indirect spin-phonon excitation.

## Results

### Generation of broad- and narrowband THz emission in NiO(001)/Pt thin films

We first address the THz-emission signal from a NiO(001)(10 nm)/Pt(2 nm) bilayer (see Methods and refs. [26,27]) when excited by 100 fs near-infrared (NIR) pulses under normal incidence (see Fig. 1a). The magnetic configuration of NiO is detailed in Supplementary Note 6. Using THz emission time-domain spectroscopy (see Methods and

Ref. [9]), we detect an emitted THz signal with two key features as shown in Fig. 1b: i) a very short THz pulse followed by ii) decaying oscillations with a periodicity of around 1 ps. In the frequency domain (inset of Fig. 1b), these two responses respectively correspond to i) a broadband contribution up to 3 THz (limited by the cutoff frequency of our detection crystal) and ii) a narrowband contribution centered at 1.1 THz.

The striking presence of oscillations at 1.1 THz is in direct agreement with the expected high-frequency magnon branch of NiO[28,29], which could only be observed previously in single crystals but not in thin films[15] due to the weak dipolar field generated by AFM magnon modes[28,30,31]. In application-relevant NiO/Pt bilayers, the spin current carried by propagating THz magnons can be converted into a transient THz charge-current through inverse spin-Hall effect in the Pt layer[15]. The presence of clear oscillations with a long lifetime of around 10 ps (compared to 30 ps in single crystals[28]), leading to the narrowband emission, indicates the low magnetic damping of the NiO thin films capped with Pt (Gilbert damping parameter $\alpha = 0.006$). In optimized thin films (see Supplementary Note 1), the combined presence of broadband and narrowband contributions indicates that different emission mechanisms may contribute to the THz signal.

We analyze the symmetries of the THz signal in Fig. 2a by measuring the dependence of the THz-emission signal in (001) oriented NiO films as a function of the in-plane angle $\theta$ i.e., the angle between the NiO [010] sample edge and the $x$ axis (see Fig. 1a). The THz signal exhibits a uniaxial dependence, and the two THz emission lobes are opposite in phase as shown in Fig. 2b. Moreover, the emission axis at $\theta = \theta_0 \simeq 35°$ coincides with the orientation of the main $T$-domain in these thin films[32]. This observation indicates a THz emission originating from regions with a constant Néel vector orientation, that are AFM monodomains of NiO with an area larger than the optical pump area (the spot is about 200 μm²) as confirmed by magneto-optic imaging[26,32] (inset of Fig. 2b and Supplementary Note 2[33]).

Figure 2c shows that the THz-emission signal is independent of the linear pump polarization angle $\alpha$ defined with respect to $y$ for the NiO(001) film. This result suggests that light absorption in the Pt is the driving mechanism of ultrafast spin-current generation, in contrast to a recent report in (111) NiO thin films[15]. One must notice that there is no light absorption in the NiO layer as the pump photon energy of 1.5 eV is below the NiO bandgap of 4 eV, as confirmed by the linear fluence dependence that excludes multi-photon processes (see Supplementary Note 3). Moreover, the THz signal does reverse in phase upon reversal of the sample or by changing the capping layer from Pt to W

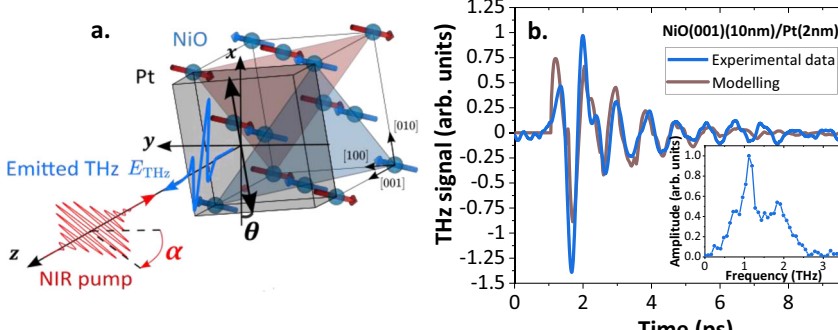

**Fig. 1 | Laser induced coherent and incoherent THz emission from NiO/Pt bilayer. a** Schematic of the setup. Femtosecond NIR laser pulses excite NiO/Pt bilayers and the THz emission is collected at normal incidence in reflection geometry and detected using electro-optic sampling. $\theta$ corresponds to the in-plane sample orientation (defined from the sample edge [010] i.e. $\theta = 0°$ corresponds to the [010] sample edge along $x$). $\alpha$ corresponds to the pump polarization angle (defined with respect to the $y$ axis where $\alpha = 0°$). In the lab frame $(x, y, z)$, $z$ is the

axis normal to the sample interface. For clarity, only the NiO magnetic sublattices without the oxygen atoms are represented. **b** Time domain THz emission from a low damping NiO(001) (10 nm)/Pt(2 nm) bilayer with the presence of oscillations about 1 ps of period (blue) and modeled THz response using magnetization dynamics simulation (brown). Inset: Fourier transform of the time domain THz signal.

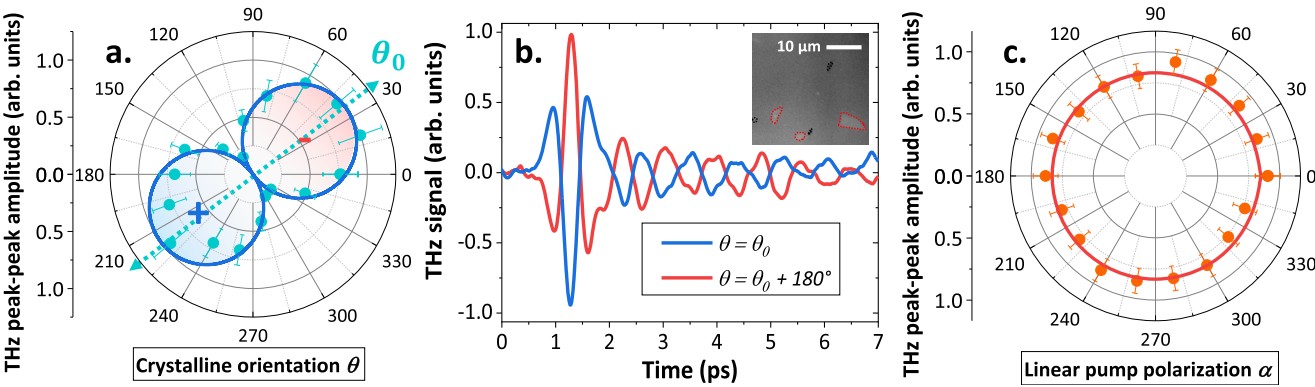

**Fig. 2 | Symmetries of the THz-emission signal in NiO(001)(10 nm)/Pt(2 nm) bilayer. a** Dependence of the THz signal on the in-plane angle $\theta$, describing a rotation of the sample around the surface normal. THz emission shows a uniaxial behavior consistent with the presence of large $T$-domains (hundreds of microns in size) contribution. **b** THz emission for $\theta = \theta_0 \simeq 35°$ and $\theta = \theta_0 + 180°$ shows a sign reversal (± label in panel a). The inset shows the magneto-optical birefringence imaging of an as grown NiO/Pt bilayer, presenting a majority of one $T$-domain orientation (gray contrast) with a small contribution of orthogonal minority $T$-domain (white contrast circled in red). The black spots are defects on the sample surface. **c** Dependence of the THz signal on the linear pump polarization (angle $\alpha$) showing an isotropic behavior in line with a thermal generation of magnons from light absorption in the Pt. Error bars correspond to standard deviations.

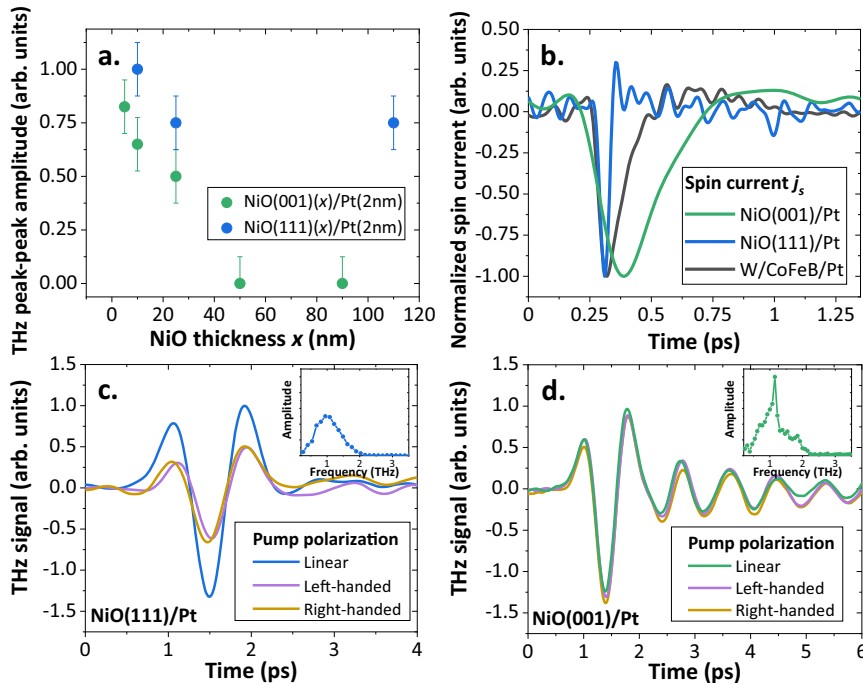

**Fig. 3 | THz emission and dynamics for (001) vs. (111) NiO thin films. a** THz peak-to-peak amplitude for NiO(001) (green) and NiO(111) (blue) samples as a function of the NiO layer thickness $x$. Error bars correspond to standard deviations. **b** Extracted spin-current $j_s(t)$ entering the Pt layer generated in NiO(001)(10 nm)/Pt(2 nm) and NiO(111)(110 nm)/Pt(10 nm) samples compared to a fully metallic W(2 nm)/CoFeB(1.8 nm)/Pt(2 nm) THz spintronic emitter. The ultrafast (<50 fs) rise and decay times of the spin-current for (111) sample are in line with off-resonant instantaneous optical-spin torque, while the longer dynamics (>300 fs) in the (001) sample are in line with a thermal excitation mediated by optical absorption and phonons. 1 THz oscillations are not visible in this figure due to the time window scale. Effect of the pump polarization (linear or circular) on the THz-emission signal for (**c**) NiO(111) (25 nm)/Pt (2 nm) and (**d**) NiO(001) (25 nm)/Pt (2 nm). Arbitrary zero-times are shifted for clarity. Insets: Fourier transform of the time domain signals.

(see Supplementary Note 4), as expected for a spin-to-charge (SCC) mechanism and not for a potential magnetic dipolar emission from the dynamics of the AFM moments. The THz-emission signal is also constant for external magnetic field of up to 200 mT (see Supplementary Note 5) in line with the large spin-flop fields in NiO (> 10 T)[27].

**Ultrafast mechanisms leading to THz magnonic spin-currents**

To identify the THz emission mechanisms in NiO thin films, we measure the THz signal as a function of the NiO thickness (ranging from 5 nm to 110 nm) for two different growth orientations: either (001), as reported above, or (111), in which optical torques were recently reported[15], as shown in Fig. 3a. For (001)-oriented films, the THz emission is maximal for thinnest NiO and decreases to zero as the NiO thickness increases. The 1 THz oscillations in the time-domain can be resolved only for the optimized 5 and 10 nm thick (001) samples (see Supplementary Note 1). The unexpected thickness dependence of the THz signal can be associated with a relaxation of the static strain in thicker NiO films[26], together with potential destructive interferences of the generated magnonic current (see Supplementary Note 6). On the contrary, NiO(111) films exhibit a sizeable THz emission even for a large

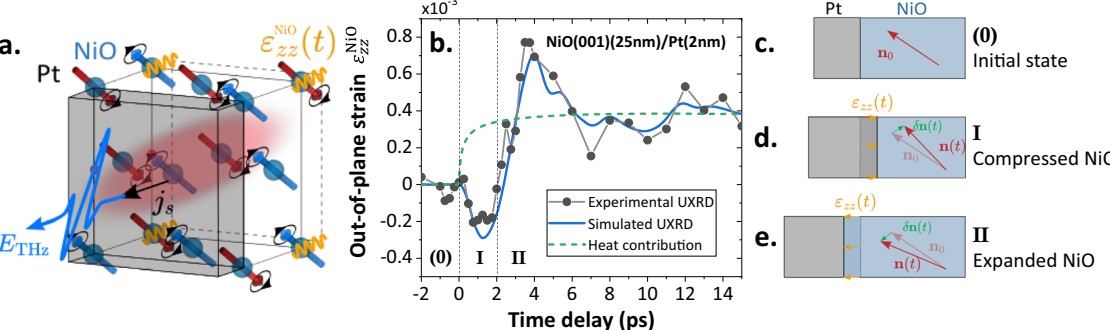

**Fig. 4 | Ultrafast strain dynamics in NiO(001)/Pt leading to the THz emission.**
**a** Sketch of the ultrafast spin-phonon interactions. The optical pump pulse is absorbed in Pt leading to an ultrafast lattice expansion of Pt following ultrafast electron-phonon coupling. A bipolar strain wave is thus launched into the NiO(001) layer (c–e). Via magneto-striction, this strain wave induces a deflection of the Néel vector (d) followed by out-of-plane oscillations, i.e. magnon excitations. A magnonic current flows into the Pt and leads to THz radiation by SCC. The temperature increase in the NiO could additionally drive the spins out of equilibrium through the temperature dependence of the anisotropy field. **b** Out-of-plane mean strain $\varepsilon_{zz}^{NiO}$ (gray) as a function of delay time mapped by ultrafast reciprocal space mapping (URSM) in a NiO(001)(25 nm)/Pt(2 nm) sample. The simulated strain (blue) includes contributions from the strain wave and from the quasi-static lattice expansion due to heating of NiO (dashed green line). The regions noted as (0), I and II correspond to the initial NiO/Pt bilayer state (**c**), the compressed NiO state (**d**) and the expanded NiO state (**e**). The red arrow represents the dynamical Néel vector, the green arrow represents the Néel vector variations and the yellow arrow represents the out-of-plane strain.

thickness of 110 nm unlike the (001) series. These observations point towards different excitation mechanisms depending on the orientation of the NiO films[15,20].

In Fig. 3c, d, we present the effect of the pump polarization (linear vs. circular) on the THz generation. Notably, we confirm the presence of two different behaviors for (001) and (111) orientations. For NiO(111) films, the THz amplitude is reduced to half the signal with circular pump polarization as shown in Fig. 3c. This hints at a direct off-resonant optical torque excitation of the NiO modes, associated in recent work to the inverse Cotton-Mouton effect (ICME)[15,20,21], which is line with the dependence on the linear pump polarization (see Supplementary Note 7). The reduction by a factor two for circular pumping is thus explained by the fact that only one of the two linear polarization components contributes to the detected THz signal. This feature is in stark contrast with the behavior observed for (001) films, where we find a THz-emission signal that is independent of the pump helicity (Figs. 2c, 3d), which we assign to light absorption in the Pt film. The polarization analysis agrees with a purely thermal origin of the emission of coherent THz magnons for the (001)-oriented films. We thus find strong evidences of two distinct excitation mechanisms, mediated either by light absorption in the Pt layer or direct off-resonant optical processes (by ICME) in the NiO layer, that can both contribute to the THz excitation of AFM insulating thin films.

Next, we explore the dynamics of these different mechanisms by extracting the temporal profile of the ultrafast spin-current using 20 fs pump pulses as illustrated in Fig. 3b (see Methods section and ref. [34] for details). The three characterized samples are: a metallic W/CoFeB/Pt THz spintronic emitter[8] (TeraSpinTec GmbH) and two NiO/Pt bilayers with (001) and (111) orientations. The spin-current profile of the metallic THz emitter shows a typical rise time about 70 fs (associated with ultrafast demagnetization)[35] and decays within 250 fs (associated with the electron-spin relaxation time in metals)[35]. The NiO(111)/Pt sample presents an even faster spin-current rise time and decay time (< 50 fs). In contrast, the spin-current in NiO(001)(10 nm)/Pt builds up over a much longer timescale of around 200–250 fs and relaxes over 300 fs. The THz spin-current dynamics further highlight the two generation mechanisms at stake in the NiO thin films. In case of i) (111) NiO orientation, off-resonant Raman-type torques acting on the magnetization are known to exhibit a quasi-instantaneous response, which explains the ultrafast rise time of the THz spin-current[20,21]. On the other hand, for ii) (001) NiO orientation, the slower generation mechanism can be explained by indirect thermal processes that require first the

absorption of photons in Pt and subsequent processes that we will discuss next[34].

## Strain-wave driven THz magnonic excitation

Finally, we investigate the origin of the indirect thermal excitation in (001) oriented NiO films by determining its spatio-temporal expansion (see Fig. 4a). We first measure the transient mean strain of the NiO layer $\varepsilon_{zz}^{NiO}(t)$ using ultrafast reciprocal space mapping (URSM, see details in Methods and Refs. [36,37]). Figure 4b presents the out-of-plane strain $\varepsilon_{zz}^{NiO}(t)$ (gray dots) derived from time-resolved reciprocal space maps (see Supplementary Note 8) around the (004) reflection peak of a NiO(001)(25 nm)/Pt(2 nm) thin film. We note that the very weak (004) peak signal prevents us from measuring thinner NiO films. The temporal strain response is characteristic of a bipolar strain wave driven by the ultrafast expansion of Pt upon optical heating, with a leading compression and a trailing expansion[23]. The strain wave is launched by the stress from electrons and phonons in Pt induced by the pump absorption by the electrons and the subsequent distribution of energy to the phonons (coupling time about 500 fs). From the initial state pictured in Fig. 4c, we observe three characteristic steps in the time-domain response: i) a compression of the NiO lattice up to 1.5–2 ps after pump pulse arrival due to the ultrafast expansion of Pt lattice upon optical heating (Fig. 4d). Then, we observe ii) a maximum expansion of NiO at 4 ps when the compressive part of the strain wave leaves NiO towards the substrate (Fig. 4e) and finally iii) a quasi-static (very slowly relaxing) expansion of NiO after 6 ps.

We then model the transient strain of NiO (blue line in Fig. 4b) using a simple one temperature model (and the Python toolbox *udkm1Dsim*[38]). We can thus obtain further insights into the dynamic of the system by calculating the spatio-temporal strain $\varepsilon_{zz}(z, t)$ and the temperature increase (see details in Supplementary Note 8). The total modeled strain in Fig. 4b (blue line) matches well the experimental results. It consists of an intrinsic heat expansion of NiO due to incoherent phonons excited by the temperature increase after heat transport from Pt into NiO (green dashed line), which is superimposed by propagating strain waves (coherent phonons) driven by Pt. It highlights two phononic contributions to the lattice response within the first picosecond: heating of NiO at the interface and compression of the first nanometers of NiO by the strain wave. From our model, we estimate the amplitude of the compression wave front about $6 \times 10^{-6}$ in the THz emission experiment for a fluence of about 10 μJ cm$^{-2}$ (1000 times smaller than in the URSM experiment). This amplitude is

sufficient to launch the precession of the out-of-plane magnon mode of NiO via magneto-strictions[16,39,40] as discussed later.

## Discussions

We finally discuss how different types of external torques $\boldsymbol{\Gamma}$ can induce the observed THz magnetization dynamics. Due to the wide spectrum of the pump laser pulse, the triggered magnetic dynamics can include a resonant excitation of THz magnon modes superimposed onto transient (non-resonant) oscillations in a wide frequency range (see Fig. 1b). We describe the magnetic state as a single-domain state of NiO with the Néel vector $\boldsymbol{n} \equiv \mathbf{M_1} - \mathbf{M_2}$ and solve the standard equations of motion:[39]

$$\boldsymbol{n_0} \times \left( \ddot{\boldsymbol{\delta n}} + 2\gamma_{\mathrm{AF}} \dot{\boldsymbol{\delta n}} - c^2 \triangle \boldsymbol{\delta n} + \omega_{\mathrm{AF}}^2(T) \boldsymbol{\delta n} \right) = \gamma^2 H_{\mathrm{ex}}(\boldsymbol{n_0} \times \boldsymbol{\Gamma}), \quad (1)$$

where $\gamma$ is the gyromagnetic ratio, $H_{\mathrm{ex}}$ is the exchange field that keeps the magnetic sublattice moments antiparallel, $c$ is the limiting magnon velocity, $\omega_{\mathrm{AF}}$ is the circular frequency of the magnetic oscillations and $\gamma_{\mathrm{AF}}$ is the damping constant. Magnons are described as small deviations $\boldsymbol{\delta n}$ of the Néel vector from the equilibrium $\boldsymbol{n_0}$. Magnon spectra of NiO include two linearly polarized magnon branches[40], a low-frequency branch (with frequency ~180 GHz) and a high-frequency branch (with frequency ~1 THz). In the high-frequency branch, $\boldsymbol{\delta n}$ oscillates out of the easy magnetic plane (either || [111] for NiO(111) and || [5 5 19] for NiO(001) due to a lattice distortion along [001][26]), leading to a time-dependent dynamic magnetization $\boldsymbol{m} = \boldsymbol{n_0} \times \boldsymbol{\delta n}/(\gamma H_{\mathrm{ex}})$ in the easy-plane. For the low-frequency branch, $\boldsymbol{\delta n}$ oscillates within the AFM plane with a dynamic magnetization $\boldsymbol{m}$ perpendicular to the AFM plane. The mode dynamics can be excited by an external torque $\boldsymbol{\Gamma}$ along $\boldsymbol{\delta n}$ multiplied by the exchange field $H_{\mathrm{ex}}$ (see Eq. (1)). A generated THz-emission signal $\boldsymbol{E}_{\mathrm{THz}}$ then emerges through the ISHE when the THz spin-current polarized along $\boldsymbol{m}$ has a finite projection onto the Pt plane. As such, only the high-frequency magnon branch can contribute to the THz generation for (111) films while both modes could contribute for (001) films.

For (111) films, the measured THz-emission signals suggest that the main origin is an optical torque due to the off-resonant direct interaction of the magnetic moments with the linearly polarized light (see Fig. 3c), as also recently observed[15]. We can exclude the inverse Faraday effect, as it requires circular (or elliptic) polarization of light and consider the ICME[21] (see Supplementary Note 6). As the off-resonant optical excitation of magnetic dynamics (such as the ICME) originates from quantum processes, the response time (spin-current rise time) of the NiO is defined by the associated linewidth around the experimental resolution of 20 fs (as seen in Fig. 3b).

For (001) films, we find a largely pump-polarization independent THz-emission and thus consider how the torques induced by optical absorption in the Pt layer can generate a non-zero spin current from the NiO into the Pt layer. First, we can exclude the standard bulk spin-Seebeck effect (SSE)[41,42] observed in ferromagnets[43], since it scales with the amplitude of the applied field for a compensated antiferromagnet[41] (see also Supplementary Note 9). We therefore consider time-dependent spin-phonon interactions with two contributions. First, a coherent strain wave as measured in Fig. 4 can trigger magnetization dynamics through the magneto-elastic effects[16,44,45], as discussed by ref. [46], which is a plausible mechanism for longitudinal out-of-plane strain along the [001] direction. Injection of an ultrafast strain wave into NiO as observed in Fig. 4 results in a bipolar modulation of the exchange interaction. Second, the lattice-heating due to incoherent phonons contributes to the dynamical strain and leads to a change of magnetic anisotropy. These effects can beare modeled by a torque $\Gamma \propto \lambda_{11} n_{0z} \varepsilon_{zz}(z, t)$, where $\lambda_{11}$ is the magneto-elastic constant that triggers the dynamic magnetization $\boldsymbol{m}(t)$ (with a deflection angle of about 0.3° for a maximum out-of-plane strain wavefront with $\varepsilon_{zz} = 6 \times 10^{-6}$, see Supplementary Note 6). This torque leads to a tilt of the easy

magnetic plane, followed by a 1 THz out-of-plane oscillations of the Néel vector (see Supplementary Note 6 for the details). On the contrary, for (111) films, the ultrafast strain $\varepsilon_{zz}(t)$ induces oscillations of the Néel vector staying within the (111) plane, and does not couple with the Néel vector components that excite the THz magnon branch (see Supplementary Note 6). This key feature explains the difference between the two orientations of the NiO films[21]. Lastly, it should be noted that the symmetry of the (001) films allows for a non-linear (interfacial) SSE[34] proportional to the temperature gradient at the NiO/Pt interface, and arising from time-correlated spin-fluctuations in the heavy metal. This contribution (reported in the ferromagnetic insulator YIG capped with Pt[34]) has the same temporal signature as the lattice heating contribution in the Pt, with a spike signal just after the pump pulse followed by a decay in time (see Supplementary Note 6). Coherent strain, in contrast, induces oscillations of the dynamic magnetization $\boldsymbol{m}$ leading to oscillations of the injected spin current $j_s$. These combined effects contribute to the NiO magnetization dynamics and spin-current injection into Pt. The fit in Fig. 1b shows that the calculated combination of these effects (strain wave and temperature increase, see Supplementary Note 6 for details) reproduces indeed qualitatively very well the experimental observations, highlighting the different roads to generate THz magnonic current in an AFM insulator through spin-phonon interactions.

We found strong evidence for two ultrafast mechanisms of magnonic excitations in the antiferromagnet NiO interfaced with a metallic Pt layer with largely different timescales. The THz emission appears via ultrafast off-resonant optical spin torque in (111)-oriented NiO samples, whilst it comes from spin-phonon interactions exciting the high-frequency magnon branch of NiO for (001) orientation. We also demonstrate by angular crystallographic THz emission mapping that the magnon generation mechanism is directly linked to the Néel vector orientation of NiO. This work opens routes towards AFM magneto-phononic and tunable THz narrowband emission.

## Methods

### Growth process

The epitaxial NiO thin films with thickness below 100 nm were grown on MgO(001) substrates by reactive magnetron sputtering using an ULVAC QAM 4 fully automated sputtering system. Sputtered NiO($x$ nm)/Pt(2 nm) bilayers with NiO thicknesses $x$ ranging from 10 to 110 nm are realized on MgO substrates with (001) and (111) orientations. NiO was deposited in a mixed atmosphere of Ar (15 sccm) and $O_2$ (0.7 sccm) at a temperature of 430 °C and 150 W. The Pt layer was subsequently deposited in-situ in a separate chamber of the same system after cooling down the sample to room temperature. The 110 nm thick (111)-oriented NiO thin film was grown on a (0001)-oriented $Al_2O_3$ substrate at 380 °C in an oxygen atmosphere of 10 µbar via pulsed-laser deposition. The growth was monitored by in-situ reflection high-energy electron diffraction. Subsequently, a 10 nm thick Pt layer was deposited on the NiO surface by electron-beam evaporation in-situ without breaking the vacuum.

### THz time domain emission spectroscopy

A Ti:Sa oscillator is used to generate 100 fs long NIR pulses centered at 820 nm. The energy per pulse is around 3 nJ with a repetition rate of 80 MHz. The initially generated NIR pulses are horizontally polarized, and the typical beam spot diameter is around 200 µm. A half-waveplate and a quarter-waveplate allow to control the polarization state of the incident NIR pulse (resp. rotation of the linear polarization angle $\alpha$ defined from $e_y$ and circular polarization). Samples are mounted on a holder, where an in-plane magnetic field is applied by permanent magnets (around 200 mT). The sample is mounted on a rotational mount which allows azimuthal rotation of the sample (angle $\theta$). The initial orientation $\theta = 0°$ corresponds to the [010] direction of the

sample. NIR pump and THz collection are at normal incidence. The THz propagation path is enclosed in a dry-nitrogen box to avoid water absorption. THz pulses are detected by electro-optic sampling with a non-linear crystal. A 500 μm thick ZnTe ⟨110⟩ crystal is used for THz detection by electro-optic sampling. For the experiments shown in Fig. 3b, the sample is excited with near-infrared femtosecond laser pulses (central wavelength of 800 nm, duration of 10 fs, energy of 1 nJ, repetition rate of 80 MHz) from a Ti:Sa laser oscillator. In all THz measurements, the NIR fluence is about 10 μJ cm$^{-2}$. The THz-emission signal is collected in the transmission geometry and detected via electro-optic sampling in either a 1 mm or a 10 μm thick ZnTe ⟨110⟩ crystal.

**Ultrafast THz spin current extraction**

In the frequency domain, the electro-optic signal $\widetilde{S}(\omega)$ and the THz electric field $\widetilde{E}(\omega)$ behind the sample are connected by a setup-specific transfer function $\widetilde{h}(\omega)$. This frequency-dependent response function includes the propagation of the THz pulse to the detector as well as the response function of the electro-optic detection process such that:

$$\widetilde{S}(\omega) = \widetilde{h}(\omega) \cdot \widetilde{E}(\omega) \qquad (2)$$

To obtain $\widetilde{h}(\omega)$, a reference emitter with a well known $\widetilde{E}_{\text{ref}}(\omega)$ was used. In more detail, a spintronic THz reference emitter (W/CoFeB/Pt) was applied for THz emission experiments in the reflection geometry, whereas, for the transmission geometry, a 50 μm GaP ⟨110⟩ crystal served as the reference. In this way, we extracted $\widetilde{h}(\omega)$ by Eq. (2), which was then used to extract $\widetilde{E}(\omega)$ for the studied sample. The spin current $\widetilde{j}_{\text{s}}(\omega)$ is related to the extracted $\widetilde{E}(\omega)$ by a generalized Ohm's law:[8]

$$\widetilde{E}(\omega) = e\widetilde{Z}(\omega)\theta_{\text{SH}}\lambda_{\text{rel}}\widetilde{j}_{\text{s}}(\omega), \qquad (3)$$

where $-e, \widetilde{Z}(\omega), \theta_{\text{SH}}, \lambda_{\text{rel}}$ denotes the electron charge, the impedance of the sample stack, the spin Hall angle, and the spin-current relaxation length of Pt, respectively. Note that we restrict ourselves to a qualitative comparison of $\widetilde{j}_{\text{s}}(\omega)$ throughout this work.

**Ultrafast reciprocal space mapping (URSM)**

URSM experiments[36] are conducted with hard X-ray probe pulses at 8 keV, which are derived from a laser-driven X-ray plasma source[37] and have a duration of approximately 200 fs and a footprint of about 300 μm. A more than three times larger area of the Pt layer is excited by p-polarized pump pulses with a duration of 100 fs at a central wavelength of 800 nm, which are incident under 25° relative to the surface normal with a fluence of 10 mJ/cm². The strain $\varepsilon_{zz} = \Delta q_z / q_z$ is the relative change of the Bragg peak position in reciprocal space.

## Data availability

The datasets generated and/or analyzed during the current study are available in the Zenodo data repository at the address https://doi.org/10.5281/zenodo.7711870.

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

## Acknowledgements

E.R., O.Gueckstock, H.J., T.Kampfrath, M.K., T.S. S., S.D. and R.L. acknowledge financial support from the Horizon 2020 Framework Pro-gram of the European Commission under FET-Open grant agreement No. 863155 (s-Nebula). R.L. acknowledges financial support from the Horizon 2020 Framework Program of the European Commission under FET-Open grant agreement No. 964931 (TSAR). H.J. and R.L. acknow-ledge financial support from the ANR project TRAPIST. M.Mattern and M.B. acknowledge financial support from the Deutsche For-schungsgemeinschaft (DFG, German Research Foundation) via No. BA 2281/11-1 Project-No. 328545488 – TRR 227, project A10. O. Gomonay, H.M., C.S., J.S. and M.K. acknowledge financial support by the German Research Foundation DFG (CRC TRR 173 SPIN + X, projects A01, A03, A11, B02, #268565370) and TopDyn. M.K. acknowledges financial support by the DAAD (Spintronics Network #57334897 and 57524834) and the Research Council of Norway (Center of Excellence 262633 "QSpin"). R.L. and M.K. acknowledge financial support from the Horizon Europe Fra-nework Program of the European Commission (HORIZON-CL4-2021-DIGITAL-EMERGING-01) under Grant Agreement 101070287 (SWAN-on-chip). O.G and J.S. acknowledge financial support by the German Research Foundation DFG from the project TRR 288 – 422213477 (project A09). T.Kampfrath and T.S.S. acknowledge financial support from the German Research Foundation (DFG) through the collaborative research center SFB TRR 227 "Ultrafast spin dynamics" (Project ID 328545488, projects A05 and B02). T.Kikkawa and E.S. acknowledge financial sup-port from JST CREST (JPMJCR20C1 and JPMJCR20T2), JSPS KAKENHI (JP19H05600, JP20H02599, JP22H05114, and JP22K18686), and the Institute for AI and Beyond of the University of Tokyo. R.R. also acknowledges support from the Grant RYC 2019-026915-I and the Pro-ject TED2021-130930B-I00 funded by the MCIN/AEI/10.13039/501100011033 and by the ESF investing in your future and the European Union NextGenerationEU/PRTR, the Xunta de Galicia (ED431F 2022/04, ED431B 2021/013, Centro Singular de Investigación de Galicia Accred-itation 2019-2022, ED431G 2019/03) and the European Union (European Regional Development Fund - ERDF).

## Author contributions

R.L. proposed the study. T.S.S., S.D., and R.L. directed the project. E.R., O.Gueckstock, M.Micica, T.S.S., T.Kampfrath, H.J., J.M., S.D., and R.L. conducted the 100 fs based THz emission spectroscopy measurements. O.Gueckstock, T.S.S., and T.Kampfrath conducted the 10 fs based THz emission spectroscopy measurements. M.Mattern and M.B. conducted ultrafast X-ray diffraction experiments. H.M., C.S., R.R., T.Kikkawa, E.S., and M.K. performed the growth of the thin (111) and (001) NiO/Pt bilayers, and S.T.B.G. and S.G. performed the growth of the thin (111) NiO/Pt sample. O.Gomonay and J.S. developed the THz magnetization dynamics model with discussions with E.R., M.B and R.L. E.R., O.Gomo-nay, M.Mattern, M.B., T.S.S., and R.L. wrote the manuscript with com-ments from all the authors. All authors contributed to the data analyses.

## Competing interests

T.S.S. and T.Kampfrath are shareholders of TeraSpinTec GmbH and T.S.S. is an employee of TeraSpinTec GmbH.
