## [Peer Review File · Nature Communications]

Reviewers' Comments:

Reviewer #1:

Remarks to the Author:

In this work the authors study the THz emission obtained when pumping a spintronic emitter that includes an antiferromagnetic (AFM) NiO film with a femtosecond infrared laser. The work builds up on a previously published work by Qiu et al, where NiO crystals grown on the (111) plane were studied. In contrast, in the work of Rongione et al. two different types of crystals are studied, one grown on (111) and another one on (001). The authors confirm the results of Qiu et al. for the (111) orientation, but find a notably different response for the (001) direction. The (001) growth direction unveils a narrowband ~ 1 THz resonant mode after excitation, as well as a broadband THz response. The main results are surprising, and the following analysis and conclusions are thorough. Particularly, the amount of data that the authors attach to the manuscript to support their analysis is very large and rich.

The excitation of AFM resonant modes and the study of their conversion into spin-currents is a major topic in the field of AFM spintronics. The generation of THz pulses via spintronic emitters with new functionality is equally an important topic. Therefore, this work will be extremely important for both the spintronic and THz communities. For this reasons, I believe the paper should be published in Nature Communications.

I have a number of comments / questions for the authors:

1. Overall I think the author's argumentation is very sound. I agree that the signals found on the (001) film are of thermal origin, since there is no pump polarization dependence. Once we assume the THz are due to heat in the Pt layer, any direct optical-rectification mechanism can be eliminated. The authors then discuss different ways in which heat could generate the THz signals, which all involve spin dynamics : magneto-acoustics, thermal changes to magnetic anisotropy, non-linear spin Seebeck effect... However, there is one important check-up the authors failed to provide : The authors compare signals from a Pt and a W film on the (111) NiO films to demonstrate the origin of the THz is due to spin currents. However, there isn't any such comparison for the (001) NiO film. I believe that if one wants to claim with absolute certainty that the measured THz in the (001) film are due to spin-currents, two heavy metals with opposite spin Hall angle should be used. Have the authors measured this, or could they do it?
2. It would be helpful if the authors explained briefly the magnetic configuration of the NiO crystals (ex : are they canted or perfectly collinear ?).
3. I do not quite understand the non-linear SSE discussion. First, the referenced Seifert et al. Nat. Comm. paper never uses the term « non-linear » SSE. Do the authors refer to the ultrafast character, to the non-equilibrium... ? Second, in the same reference, to have a SSE a magnetization is required. In the case of NiO, M is zero unless canting is present. The authors should explain more clearly where the SSE effect would arise from.
4. Minor grammar error on line 51 : « ... modes have been predicted ... »

Jon Gorchon

Reviewer #2:

None

Reviewer #3:

Remarks to the Author:

The main result of the article of E. Rongione et al is the observation of the THz emission from optically excited NiO/Pt sample with two NiO orientations. I should note that the mere fact of THz emission measurements from an antiferromagnetic material is not anymore novel and thus the

article represents a further development of this (quite hot) topic.

In the present article the authors argue that the novelty resides in a simultaneous generation of two excitation processes with different timescales. The observed excitation (Fig. 1b) consists of a sharp peak at 1 THz and a broad response around 3THz. Here already I would argue that the presented spectrum could also be a not so sharp peak at 1THz and a broader peak at ca. 2THz, thus the deconvolution of the THz response in a peak and a broadband response should be presented.

The authors argue that the first peak comes from direct optical excitation by optical spin torque in NiO while the broadband response comes from thermal excitation of phonons in the Pt layer. If I understood correctly, they conclude that they observe the first mechanism in Ni(111)/Pt sample while the second one – in Ni(001)/Pt sample. At least this seems what is discussed in the conclusions. If this is so, then I am confused: why Ni(001) in Fig. 1b presents two peaks?. Furthermore, what are the Fourier transforms of the signals presented in Fig.3c and d? Do they also present two contributions or only one of them?

I am also confused by the difference between linear and circular polarization's for NiO(111). Why would linear polarization have a larger signal than the circular one? This is not discussed.

The experiment is supplied by two models which in the main text seem independent. They are quite simple and speculative in the sense that the relation to the experiment is discussed in general words. Other than that, the main text does not explain which of the two models are fitted in Fig.1b. Why not to Fig.3c and d? This seems as the main result from the modelling but what exactly is the model should be schematically presented in Methods.

The first model considers the time-dependence of the strain in a very simple one-dimensional model. In the main text, it does not seem to be connected to spins, the input is in temperature (how would the excitation be translated into the temperature?) For me it seems that this does not explain the THz spectrum but only shows some strain dynamics which is of course very natural in this model.

The second model is based on standard equation for AFM dynamics (1) which would explain the peak at 1THz. It also speculates about the differences between excitations in 111 and 100 directions. It is not clear if the two models are connected and how. The main text does not specify if the strain was taken from the strain model and input into the spins. If this is so then the statement that the change of exchange interactions can be modelled the same way as the change of strain and would produce the same torque would be confusing sine this would mean that the latter effect was simply not taken into account which should be specified like that.

I would suggest that the method section clearly but briefly explains the Model.

One of the most important question from the modelling should be what is the magnitude of this torque which produces the measured THz excitation with the correct timescale? Was the excitation timescale taken into account and what is the result of the spin current dynamics?

I am also confused by the sentence ·"On the contrary, for (111) films, the ultrafast strain induces oscillations of the Néel vector staying within the (111) plane and thus no spin current propagates towards the Pt layer." Don't the authors say that they have ISHE in all cases?

Please, check also the sentence in the introduction: " In this regard, AFM materials with their characteristic THz resonant modes could operate have been predicted to enable the development of narrowband THz spintronic emitters"

My conclusion is that this work should be considered as incremental in the topic of THz emission by optical excitation. I do not see that it introduces something novel rather than discusses that the two known mechanisms act simultaneously. From my point of view, the most interesting result is the ultrafast timescale of the optical spin-torque excitation (if the mechanism is really this one). However, the identification is rather speculative and the theory does not answer the question of the timescale.

Reviewer #4:
None

We thank the two reviewers for their evaluation of our work. We thank referee #1 for his positive evaluation of our work, and for their important question on the presence of a phase reversal of the inverse spin-Hall voltage for (001) NiO thin films when reversing for Pt to W (or Ta) capping. It has required a strong effort from the material growth side to get good quality deposition of W and Ta layers on optimized NiO(001) thin films but the new results are fully in line with expectations as described below. We also clarified our messages in order to evidence more clearly the novelty of our work and answer the interrogations from Referee #3.

Below we answer their different questions and remarks point-by point (in blue color, and where the modified text referings are in red color):

Comments from Referee #1

In this work the authors study the THz emission obtained when pumping a spintronic emitter that includes an antiferromagnetic (AFM) NiO film with a femtosecond infrared laser. The work builds up on a previously published work by Qiu et al, where NiO crystals grown on the (111) plane were studied. In contrast, in the work of Rongione et al. two different types of crystals are studied, one grown on (111) and another one on (001). The authors confirm the results of Qiu et al. for the (111) orientation, but find a notably different response for the (001) direction. The (001) growth direction unveils a narrowband ~1THz resonant mode after excitation, as well as a broadband THz response. The main results are surprising, and the following analysis and conclusions are thorough. Particularly, the amount of data that the authors attach to the manuscript to support their analysis is very large and rich. The excitation of AFM resonant modes and the study of their conversion into spin-currents is a major topic in the field of AFM spintronics. The generation of THz pulses via spintronic emitters with new functionality is equally an important topic. Therefore, this work will be extremely important for both the spintronic and THz communities. For this reason, I believe the paper should be published in Nature Communications.

We thank the referee for highlighting the strong interest of both spintronic and THz communities to this work. We also acknowledge him/her for emphasizing on the throughout study we have conducted on NiO/Pt bilayers, evidencing different spin-current excitation mechanisms respectively in (001) and (111) oriented thin films. As emphasized by the referee, the (001) spin current generation mechanism is notably different from previously studied (111) orientation. We answer his/her comments detailed below.

I have a number of comments / questions for the authors:

1. Overall I think the author's argumentation is very sound. I agree that the signals found on the (001) film are of thermal origin, since there is no pump polarization dependence. Once we assume the THz are due to heat in the Pt layer, any direct optical-rectification mechanism can be eliminated. The authors then discuss different ways in which heat could generate the THz signals, which all involve spin dynamics : magneto-acoustics, thermal changes to magnetic anisotropy, non-linear spin Seebeck effect... However, there is one important check-up the authors failed to provide : The authors compare signals from a Pt and a W film on the (111) NiO films to demonstrate the origin of the THz is due to spin currents. However, there isn't any such comparison for the (001) NiO film. I believe that if one wants to claim with absolute certainty that the measured THz in the (001) film are due to spin-currents, two heavy metals with opposite spin Hall angle should be used. Have the authors measured this, or could they do it?

We thank the referee for raising this particular and important point. Following his/her comment, we put an extensive effort in growing new (001) NiO samples with different heavy metal capping. In particular, we grew NiO(001)/W and NiO(001)/Ta samples, which presents a negative spin Hall angle compared to Pt and from various sources (growth performed by C. Schmitt & H. Meer from JGU, and

by T. Kikkawa, University of Tokyo). The samples are capped with AlOx to prevent oxidation of the heavy metal.

We present in Fig. R1 the THz measurements between NiO(001)(10nm)/W(2nm)/AlOx(3nm), NiO(001)(10nm)/Ta(2nm)/AlOx(2nm) compared to NiO(001)(10nm)/Pt(2nm). The measurements consisted of acquiring the THz signals from the bilayers by carefully setting the same azimuthal crystalline orientation θ on a same sample to recover the same orientation of the Néel vector. We measure a negatively phase THz signals from NiO/W and NiO/Ta bilayers compared to NiO/Pt. The smaller amplitude obtained in NiO/W and NiO/Ta arises potentially from i) a different spin-mixing conductance and/or ii) a smaller value of the spin Hall angle. Following the referee's comment, we thus demonstrate that the THz emission mechanism arise from spin-currents in NiO(001)/HM bilayers and one can notice here despite the lower signal amplitude the presence of the oscillations also in the W case.

Fig. R1 – Role of the spin Hall angle in (001) oriented NiO. (a) Comparison between NiO/Pt vs NiO/W and (b) NiO/Pt vs NiO/Ta.

Therefore, we have added the following part in the revised Suppl. Mat. S4:

“Fig. S4a presents the THz emission from NiO/Pt, and NiO/W and NiO/Ta bilayers for (001) thin films. The presence of a phase reversal of the generated THz signal for Pt (Fig. S4a-b), and W (Fig. S4a) and Ta (Fig. S4b) based bilayers, indicates that the spin-charge conversion (SCC) occurring via ISHE is responsible for the detected THz emission according to $j_c \propto \theta_{SHE}(j_s \times m)$ and given the opposite spin-Hall angle between Pt, and W and Ta. The smaller amplitude obtained in NiO/W and NiO/Ta arises potentially from i) a different spin-mixing conductance and/or ii) a smaller value of the spin Hall angle. One can nevertheless notice here that the presence of the oscillations also in the W case despite the lower signal amplitude.”

Fig. S4. Identification of ISHE as the emission mechanism in (001) thin films. (a-b) Role of the spin Hall angle in (001) oriented NiO. Comparison between (a) NiO/Pt vs NiO/W and (b) NiO/Pt vs NiO/Ta. (c) Emitted THz signal when reversing the sample surface facing the optical pump (i.e. pumping first from Pt or from the MgO substrate side). The phase reversal is in line with spin-charge conversion in Pt. In the case of pumping from the MgO substrate side, the signal has been shifted in time for clarity.

2. It would be helpful if the authors explained briefly the magnetic configuration of the NiO crystals (ex : are they canted or perfectly collinear ?).

We thank the referee for his question. We have presented in the SM6 extensive information about the NiO magnetic configuration : “NiO is an easy-plane antiferromagnet with the Néel temperature of 523 K (Ref.[1]). Below the Néel temperature, the magnetic spins lie in (111) planes where M_1 and M_2 describe opposite spin ordering between adjacent planes in equilibrium state. Magnetic structure can be realized in form of four possible T domains that are distinguished by orientation of easy magnetic planes[2]. Orientation of the magnetic moments in easy magnetic plane and T-domain structure depend on the film orientation. In NiO(111) samples, the growth conditions favor formation of a single T domain with $M_1 \uparrow \downarrow M_2$ aligned along one of the three equivalent $[11\bar{2}]$ directions in (111) plane, thus forming three equivalent S domains. NiO(001) films show multidomain structure with all four possible T domains. However, pronounced out-of-plane deformation[3] removes S domains and stabilizes the single equilibrium orientation along $[5\ 5\ \bar{1}9]$ within each of T domain. We thus consider this initial AFM state in our theoretical modelling.” **We also mention in Suppl. Mat. S5 experimental measurements (XMCD and XMLD) from C. Schmitt et al.[3] that have been performed in our NiO thin films in order to confirm the antiferromagnetic character.**

We thus added in the Main the text:

“The magnetic configuration of NiO is detailed in SM6.”

3. I do not quite understand the non-linear SSE discussion. First, the referenced Seifert et al. Nat. Comm. paper never uses the term « non-linear » SSE. Do the authors refer to the ultrafast character, to the non-equilibrium... ? Second, in the same reference, to have a SSE a magnetization is required. In the case of NiO, M is zero unless canting is present. The authors should explain more clearly where the SSE effect would arise from.

We thank referee for this comment. We introduced the term “nonlinear” SSE in the present paper in order to distinguish the observed effect from the “standard” SSE observed in ferromagnets and AFs in presence of the external magnetic field. In case of a “standard” SSE a temperature gradient induces the magnon flux. If magnons are spin-polarized (as in FM) or there is imbalance between spin-up and spin-down magnons (as in AF in presence of the magnetic field), magnon flux is spin-

polarized and creates a spin current. In this scenario the value of spin current is proportional to the average spin of the magnetic fluctuations (magnons) and in this sense this effect is linear.

In the present paper we consider more complicated mechanism in which the average spin of magnons is zero (AF in absence of the external magnetic field). In this case, the presence of the Pt layer is important and the Pt electrode plays not only the role of heater, but also is considered as a source of spin fluctuations (with average zero spin). Similar to the model of Seifert et al.[4], we assume that spin fluctuations in Pt are time-correlated, and that Pt spins interact with NiO and create a torque on local magnetic moments. First, Pt spins induce rotation of NiO moments, the second Pt spins with the same polarization that kick the system short after the first one, creates spin-polarized magnons in AFs. This effect appears as a second order effect, is proportional to spin-spin correlations and we called it "nonlinear". May be this term could be confusing. However, the term SSE in this situation is also not fully appropriate, as compared to "linear", or standard SEE, temperature gradient is not enough, it is necessary to have additional degree of freedom (free Pt spins).

To avoid any misunderstanding regarding previous works in the literature, we thus refer to this term as "standard" spin-Seebeck in the manuscript and clarify the different spin-Seebeck contributions:

In the main text:

"First, we can exclude the standard spin-Seebeck effect (SSE) [5], [6] observed in ferromagnets [4], since it scales with the amplitude of the applied field for a compensated antiferromagnet [5] (see also SM9)."

In Suppl. Mat. S9:

*"**Spin-Seebeck effect contributions (SSE).** First, it should be noted that the standard spin-Seebeck effect, present in magnetic systems for which a magnon flux carries angular momentum (ferromagnets, easy-axis antiferromagnets under applied field), seems to be irrelevant in the NiO/Pt system. A generated thermal gradient in the NiO (hot interface with Pt and cold interface with MgO) is a necessary factor but not the only element that needs to be considered for the spin-Seebeck effect. In NiO, although it possesses a sizeable macroscopic AFM ordering[7], the two magnon branches are non-degenerate at zero magnetic field[5], [6], which excludes the spin-Seebeck excitation mechanism for building a net spin current. Moreover, the weak temperature dependence of the NiO(001)(10nm)/Pt(2nm) THz emission (see SM9) does not fit the theoretical expectations for SSE[5].*

It should then be noted that the symmetry of (001) films allows for a second spin-Seebeck effect proportional to the temperature gradient at the NiO/Pt interface and to spin-spin correlations, with the same symmetry as the thermo-magneto-elastic effect. In order to describe this contribution, we follow the approach of Ref.[4] used for YIG/Pt systems and consider a torque induced by spin fluctuations s_{Pt} in the Pt layer:

4. Minor grammar error on line 51 : « ... modes have been predicted ... »

We thank the referee for his/her careful review and the error is now corrected in the text. We have modified it accordingly:

"In this regard, AFM materials with their characteristic THz resonant modes have been predicted to enable the development of narrowband THz spintronic emitters, i.e. THz nano-oscillators, which can be driven by spin-orbit torques [14]."

Comments from Reviewer #3

The main result of the article of E. Rongione et al is the observation of the THz emission from optically excited NiO/Pt sample with two NiO orientations. I should note that the mere fact of THz emission measurements from an antiferromagnetic material is not anymore novel and thus the article represents a further development of this (quite hot) topic.

In the present article the authors argue that the novelty resides in a simultaneous generation of two excitation processes with different timescales. The observed excitation (Fig. 1b) consists of a sharp peak at 1 THz and a broad response around 3THz. Here already I would argue that the presented spectrum could also be a not so sharp peak at 1THz and a broader peak at ca. 2THz, thus the deconvolution of the THz response in a peak and a broadband response should be presented.

We thank the reviewer for their comments. While reviewer #1 recommends publication, the reviewer #3 has asked to clarify the novelty. We want to clarify below and in the revised manuscript a few points that were potentially not clear in the initial version. We first want to point out that the main novelty of our work does not only rely on the first observation of antiferromagnetic magnon mode in the THz range in an antiferromagnetic thin films using spin-to-charge conversion process. We also observe a new excitation mechanism of THz dynamics in antiferromagnets that is not due to direct optical excitations like the standard Cotton-Mouton effect (that we also observe in (111) thin films) but to indirect ultra-fast thermal excitations in (001) oriented thin films. We then want to clarify to the referee that the excitation process is not simultaneous for a specific orientation, but that the excitation process depends on the symmetries of the NiO orientation (either (001) or (111)), and so do the timescales. This new mechanism in (001) thin films is thus confirmed by looking at the symmetry of the excitation process (Fig. 3d), at its timescale (Fig .3d), at the ultrafast strain response (Fig. 4), and represent we think a new and promising approach to develop optomagnonics device and THz magneto-acoustic devices.

Following the comment of the referee on the origin of the 1 THz peak, we present in Fig. R2 a deconvolution of the Fourier transform with two narrowband contributions centred around 1.1 THz and 1.9 THz, and the broadband response. The extracted full width at half maximum (FWHM) of the two narrowband contributions are around 0.15 THz, which is limited by the frequency resolution of our detection and thus cannot allow us to get a damping extraction. Indeed, the use of an electro-optic crystal of finite size for THz detection introduces a measurable THz echo limiting the frequency resolution at $\Delta f \simeq c/2d_{cr}n_{cr}$ (d_{cr} and n_{cr} are respectively the crystal thickness and THz refractive index), thus typically of values around $\Delta f \simeq 0.1$ THz for a 1 mm-thick ZnTe crystal (of refractive index around $n_{cr}=3$ [9]). Beyond these considerations, we want to point out that an alternative approach to identify the narrowband vs the broadband contribution consists in comparing the signal between different thin films as shown in Suppl. Mat. S1.

Fig. R2 – Deconvolution of the measured THz spectra from NiO(001)(10nm)/Pt(2nm) into a broadband response (green) and two narrowband contributions (red at 1.1 and 1.9 THz).

The authors argue that the first peak comes from direct optical excitation by optical spin torque in NiO while the broadband response comes from thermal excitation of phonons in the Pt layer. If I understood correctly, they conclude that they observe the first mechanism in Ni(111)/Pt sample while the second one – in Ni(001)/Pt sample. At least this seems what is discussed in the conclusions. If this is so, then I am confused: why Ni(001) in Fig. 1b presents two peaks? Furthermore, what are the Fourier transforms of the signals presented in Fig.3c and d? Do they also present two contributions or only one of them?

One sentence from our conclusion was possibly misleading, and we want to clarify here our observations described in the main text. We do not claim that “the first peak comes from direct optical excitation by optical spin torque while the broadband response comes from thermal excitation of phonons in the Pt layer”. What we presented in this study is that two separate excitation mechanisms occur depending on the NiO orientation (either (001) and (111)). The ultrafast spin-phonon interactions leading to the THz emission is the main effect happening in NiO(001)/Pt, while the NiO(111)/Pt generation mechanism is mainly via off-resonant Raman-like spin-optical torque (the latter mechanism being demonstrated previously [3], [4]). As mentioned above these different mechanisms have different symmetries (Fig. 3c-d, see also Fig. 2 versus Fig. S10), and time scales (Fig. 3b) .

We thus change the following sentence in our conclusion:

“The THz emission appears via ultrafast off-resonant optical spin torque in (111)-oriented NiO samples, whilst it comes from spin-phonon interactions exciting the high-frequency magnon branch of NiO for (001) orientation.”

Regarding the presence of 1 THz oscillations, we observe these only in optimized (001) thin films capped with either Pt (or W as presented in the revised manuscript following the comment from referee 1) and not in NiO(111) thin films (or in unoptimized (001) thin films as shown in Fig. S1). The narrowband peak in the THz spectra of NiO(001)/Pt corresponds to the response of the Néel vector excitation. On the contrary, the broadband contribution is present in all NiO thin films, both (001) and (111), and can be associated to the excitation of an uncoherent magnon bath (in analogy with

observations in YIG/Pt[4] or in the recent arXiv[12]). To clarify this point, we present in Fig. R3 the Fourier transform of the THz signal from NiO(111)/Pt and NiO(001)/Pt (respectively from Fig. 3c and 3d of the main text). We clearly see that the two narrowband contributions appear as complementary to the broadband response in the case of NiO(001)/Pt while such contributions could not be resolved in NiO(111)/Pt.

Fig. R3 – Comparison of the THz spectra (Fourier transform) obtained from NiO(111)/Pt (blue) and NiO(001)/Pt (green). The latter presents narrowband contributions overprinted on a broadband response, similar to the one derived from NiO(111)/Pt emission. The broadband responses are normalized at the same level for clarity.

For clarity, we have added the Fourier transform of this signal in the inset of Fig. 3c-d:

Fig. 3. THz emission and dynamics for (001) vs. (111) NiO thin films. (a) THz peak-to-peak amplitude for NiO(001) (green) and NiO(111) (blue) samples as a function of the NiO layer thickness x . (b) Extracted spin-current $j_s(t)$ entering the Pt layer generated in NiO(001)(10nm)/Pt(2nm) and NiO(111)(110nm)/Pt(10nm) samples compared to a fully metallic W(2nm)/CoFeB(1.8nm)/Pt(2nm) THz spintronic emitter. The ultrafast (<50 fs) rise and decay time of the spin-current for (111) sample is in line with inverse Cotton-Mouton effect mediated excitation while longer dynamics (>300 fs) in the (001) sample is in line with a thermal excitation mediated by optical absorption and phonons. 1 THz oscillations are not visible in this figure due to the time window scale. (c-d) Effect of the pump polarization (linear or circular) on the THz-emission signal for (c) NiO(111) (25 nm)/Pt(2nm) and (d) NiO(001) (25 nm)/Pt (2 nm). Arbitrary zero-times are shifted for clarity. *Insets: Fourier transform of the time domain signals.*

I am also confused by the difference between linear and circular polarization's for NiO(111). Why would linear polarization have a larger signal than the circular one? This is not discussed.

For the pump polarization-dependent THz emission NiO(111) oriented thin films, our experiments point towards a generation mechanism via an instantaneous off-resonant optical-spin torque, similar to the inverse Cotton-Mouton effect. In this view, the reduced THz amplitude found for the circular pump polarization arises from the fact that pump electric field can be written as $E_{pump}^{circ} = \frac{1}{\sqrt{2}}(E_{lin}^s + iE_{lin}^p)$. Thus, as we only map the THz component polarized along the horizontal plane e_x (i.e the p -polarized THz component), we measure a THz signal which is half of the THz signal recovered in the case of a fully horizontally polarized pump. These observations on the symmetry of the inverse Cotton-Mouton effect in NiO(111) are in line with previous observations in NiO (Refs.[10], [11], [13]).

We added the following part in the SM7:

"In this view, the reduced THz amplitude found for the circular pump polarization arises from the fact that pump electric field can be written as $E_{pump}^{circ} = \frac{1}{\sqrt{2}}(E_{lin}^p + iE_{lin}^s)$. Thus, as we only map the p -polarized THz component, we measure a THz signal which is half the amplitude recovered for a fully p -polarized pump."

And the following sentence in the main text:

"The reduction by a factor two for circular pumping is thus explained by the fact that only one of the two linear polarization components contributes to the detected THz signal."

The experiment is supplied by two models which in the main text seem independent. They are quite simple and speculative in the sense that the relation to the experiment is discussed in general words. Other than that, the main text does not explain which of the two models are fitted in Fig.1b. Why not to Fig.3c and d? This seems as the main result from the modelling but what exactly is the model should be schematically presented in Methods.

The first model considers the time-dependence of the strain in a very simple one-dimensional model. In the main text, it does not seem to be connected to spins, the input is in temperature (how would the excitation be translated into the temperature?) For me it seems that this does not explain the THz spectrum but only shows some strain dynamics which is of course very natural in this model. The second model is based on standard equation for AFM dynamics (1) which would explain the peak at 1THz. It also speculates about the differences between excitations in 111 and 100 directions. It is not clear if the two models are connected and how.

We thank the referee for this valid question. First, regarding the response in (111) thin films (Fig. 3c), it can be simply described in term of off-resonant optical spin torque excitation as explained in Suppl. Mat. S6 (Section “Off-resonant optical spin torque via Inverse Cotton-Mouton effect (ICME)”). We do not developed this case in extensive details as it has been reported previously in the literature, including the recent work from Qiu et al.[10] and only a broadband THz response is observed in this case in line with these previous observations.

Regarding the (001) thin films, we model the magnetization dynamics of NiO using the equation of motion described in Suppl. Mat. S6 (Section “Equation of motion of the antiferromagnetic order”) (Eq. S1 and Eq. 1 of the Main text), by which we analyse the different torques that can excite THz mode responses. It can be used as input parameters for ultrafast magneto-striction induced by dynamical strain (as described in Suppl. Mat. S6 – Section “Thermo-magneto-elastic effect”) or spin-Seebeck effects (as described in Suppl. Mat. S6 – Section “Spin-Seebeck effect contributions (SSE)”). The ultrafast strain and temperature profiles are extracted from the UXRd experiments and from their modelling as described in Suppl. Mat S8. The result of this modelling (called “first model” by the referee) is thus used as input parameters in the Eq. S1 to calculate the magnetization response (called “second model” by the referee). We clarified this point in the revised version.

The final result of the magnetization modelling is then the same for Fig. 3 and Fig. 1, which are similar samples with similar NiO and Pt properties. Beyond these methodology aspects, we discuss and answer below more precise questions regarding the modelling.

The main text does not specify if the strain was taken from the strain model and input into the spins. If this is so then the statement that the change of exchange interactions can be modelled the same way as the change of strain and would produce the same torque would be confusing sine this would mean that the latter effect was simply not taken into account which should be specified like that. I would suggest that the method section clearly but briefly explains the Model.

Regarding the strain response, we want to point that it is first measured experimentally as shown by the experimental data points of UXRd shown in Fig. 4.b. The modelling presented in Suppl. Mat S8 is then used to understand the origin of this ultra-fast strain response. The response of the modelling can be decomposed into a coherent strain wave and uncoherent phonons (leading to thermal expansion). This full strain response is then used as input parameters when modelling the equations of motion for the Néel vector (Eq. (1) of the main text) triggered by different types of torques.

In case of the thermo-magneto-elastic effect, the torque is induced by the out-of-plane component of the strain through magnetoelastic coupling (Eq. (S7)). The microscopic origin of the corresponding component of magnetoelastic tensor λ_{11} is related with space dependence of the exchange interactions and in this way the change of exchange interactions is automatically included into the model. In (001) geometry the out-of-plane component of strain $\varepsilon_{zz}(z, t)$ couples with the component of the Néel vector that is responsible for excitation of THz mode. This strain appears due to the thermal expansion of Pt layer after exposing of laser pulse and consequent elastic response of NiO layer. For simulation of the response (Fig. 3) we solved numerically the coupled system of i) elasticity equation (Eq. S12), ii) Fourier equation for thermo-conductivity; iii) dynamics equation for the Néel vector (1) with the torque given by Eq. (S7). As the input data (initial conditions), we used the time-profile of the laser pulse. The output signal (Fig.4) is calculated as a dynamic magnetization that results from the magnetic dynamics (in-line Eq. $m = n_0 \times \delta n / (\gamma H_{ex})$ in the main text). The presence of THz mode was checked by Fourier transform of the calculated response. The parameters

of the model are given in Table S1. The details of modelling and correspondence to the dynamics UXR data are described in SM.

Finally, the thermo-magneto-elastic mechanism is excluded for (111) geometry because the diagonal strain components $\varepsilon_{xx}(z, t)$, $\varepsilon_{yy}(z, t)$, $\varepsilon_{zz}(z, t)$ do not couple with the component of the Néel vector that is responsible for excitation of THz mode. Nondiagonal shear strains ε_{zx} , ε_{zy} that couple with THz mode in this case could be excluded due to the in-plane symmetry of the laser beam and, mainly, due to strong clamping.

To clarify these points, we added the following part in the Main text:

*“On the contrary, for (111) films, the ultrafast strain $\varepsilon_{zz}(t)$ induces oscillations of the Néel vector staying within the (111) plane, and does not couple with the Néel vector components that excite the THz magnon branch (see **SM6**).”*

And in the Suppl. Mat. :

SM6: *“We use dynamical X-ray diffraction theory to calculate a NiO Bragg peak from the modelled spatio-temporal strain in NiO. The results of this modelling correspond to the modelled transient mean strain of NiO represented as a blue line in **Fig. 4b**, which is in line with the experimental data points obtained by UXR (from the time-dependent shift of the Bragg peak in reciprocal space).”*

SM8: *“We then model the magnetization dynamics of the NiO triggered by the torque contributions associated with non-linear SSE and thermo-magneto-elastic effect in the (001) films using the Eq. (S1). To model the THz response, we use as inputs parameters the temperature and strain profile extracted by fitting the URSM data using the strain model presented in **SM7**. We used the strain profile $\varepsilon_{zz}(z, t)$ to calculate the thermo-magnetoelastic torque (S10) in NiO layer and plug it into dynamic Eq. (S4) to calculate the corresponding magnetization $\mathbf{m} = \mathbf{n}_0 \times \delta \mathbf{n} / (\gamma H_{ex})$ of the several near-interface NiO layers that contribute into the optical response E_{MAS} . We separately calculated the non-linear spin-Seebeck effect using the temperature profile extracted from the modelling in **SM7**.”*

One of the most important question from the modelling should be what is the magnitude of this torque which produces the measured THz excitation with the correct timescale? Was the excitation timescale taken into account and what is the result of the spin current dynamics?

The results of the spin current dynamics are shown in Fig. 1b: time dependence of the dynamic magnetization that induces ISHE calculated from solving coupled equations of elasticity, thermal conductivity and magnetic dynamics. As an input for calculation, we used the time and space dependence of the strain modelling.

As the THz signal cannot be quantitatively related to the opening angle of the magnetization, we cannot go beyond a qualitative agreement between the THz response and the modelling. However, we can estimate, from the amplitude of the strain response measured by UXR and from the magneto-striction coefficient of NiO, the dynamic magnetization to have an opening angle of about 0.3° (extraction explained in Suppl. Mat. S6 – Section Thermo-magneto-elastic effect) which is of the order of the opening angle measured in standard ferromagnetic resonance measurements.

Finally, we used the time scale of the strain response (From Fig. 4b) to model the magnetization dynamics and the output timescales are fully in line with the observed THz signal (from Fig. 1b).

We clarified this point in the revised version by adding in the Main text:

*“These combined effects can be modelled by a torque $\Gamma \propto \lambda_{11}n_{0z}\varepsilon_{zz}(z,t)$, where λ_{11} is the magneto-elastic constant that triggers the dynamic magnetization $\mathbf{m}(t)$ (with a deflection angle of about 0.3° for a maximum out-of-plane strain wavefront with $\varepsilon_{zz} = 6 \times 10^{-6}$, see **SM6**).”*

And in the Suppl. Mat. S6:

*“One must notice that the timescales of the modelled magnetization response are in line with the experimental THz response (**Fig. 4b**) and with the modelled ultrafast strain and temperature responses (**Fig. 4b** and **SM8**).”*

I am also confused by the sentence “On the contrary, for (111) films, the ultrafast strain induces oscillations of the Néel vector staying within the (111) plane and thus no spin current propagates towards the Pt layer.” Don’t the authors say that they have ISHE in all cases?

We thank the referee for pointing out this element. As mentioned above, with this phrase we thus mean that the thermo-magneto-elastic effect cannot induce oscillations of THz mode in (111) geometry, but this does not exclude other mechanisms, e.g., magneto-optic effect that we effectively measured in (111) NiO thin films as shown in Fig. 3c of the main text.

To clarify this point, we added the following part in the Suppl. Mat.:

*“On the contrary, for (111) films, the ultrafast strain $\varepsilon_{zz}(t)$ induces oscillations of the Néel vector staying within the (111) plane, and does not couple with the Néel vector components that excite the THz magnon branch (see **SM6**).”*

Please, check also the sentence in the introduction: “ In this regard, AFM materials with their characteristic THz resonant modes could operate have been predicted to enable the development of narrowband THz spintronic emitters”

We thank the referee for his/her careful review, we revised this sentence accordingly as following:

“In this regard, AFM materials with their characteristic THz resonant modes have been predicted to enable the development of narrowband THz spintronic emitters, i.e. THz nano-oscillators, which can be driven by spin-orbit torques [14]”

My conclusion is that this work should be considered as incremental in the topic of THz emission by optical excitation. I do not see that it introduces something novel rather than discusses that the two known mechanisms act simultaneously. From my point of view, the most interesting result is the ultrafast timescale of the optical spin-torque excitation (if the mechanism is really this one). However, the identification is rather speculative and the theory does not answer the question of the timescale.

Regarding the final comments of the reviewers, we want to emphasize here that narrowband THz emission from spintronic heterostructures have never been reported in the past and our work represent an important step forward in the understanding and development of functionalized THz spintronic devices based on antiferromagnetic materials. Whilst we also agree that the ultrafast timescale extraction is a powerful result to deepen our comprehension of ultrafast magnetization dynamics (such as the one triggered by the Inverse Cotton-Mouton effect) and on spin-current generation and relaxation in spintronic heterostructures, we want to point out here again that our study on (001) oriented NiO evidences a new approach to generate ultrafast magnetization dynamics based on spin-phonon interactions. Such an approach, related to magnetostriction launched by

ultrafast strain waves, was never demonstrated before on the ultrafast timescales and as an active method to control the resulting THz emission. We believe that our work will thus open new perspectives for the research fields of ultra-fast and antiferromagnetic spintronic.

We hope that our above responses have permitted to answer the reviewer questions satisfactorily and clarify the different points that were unclear, so that we could further evidence the key novelties of our work, and that he/she will share our opinion that our strongly revised work is suitable for publication in Nature Communications.

References used in this answer to the referee's comments.

- [1] W. L. Roth, "Neutron and Optical Studies of Domains in NiO," *Journal of Applied Physics*, vol. 31, no. 11, pp. 2000–2011, Nov. 1960, doi: 10.1063/1.1735486.
- [2] I. Sanger, V. V. Pavlov, M. Bayer, and M. Fiebig, "Distribution of antiferromagnetic spin and twin domains in NiO," *Phys. Rev. B*, vol. 74, no. 14, p. 144401, Oct. 2006, doi: 10.1103/PhysRevB.74.144401.
- [3] C. Schmitt *et al.*, "Identification of Neel Vector Orientation in Antiferromagnetic Domains Switched by Currents in NiO/Pt Thin Films," *Phys. Rev. Applied*, vol. 15, no. 3, p. 034047, Mar. 2021, doi: 10.1103/PhysRevApplied.15.034047.
- [4] T. S. Seifert *et al.*, "Femtosecond formation dynamics of the spin Seebeck effect revealed by terahertz spectroscopy," *Nat Commun*, vol. 9, no. 1, p. 2899, Dec. 2018, doi: 10.1038/s41467-018-05135-2.
- [5] S. M. Rezende, A. Azevedo, and R. L. Rodriguez-Suarez, "Magnon diffusion theory for the spin Seebeck effect in ferromagnetic and antiferromagnetic insulators," *J. Phys. D: Appl. Phys.*, vol. 51, no. 17, p. 174004, May 2018, doi: 10.1088/1361-6463/aab5f8.
- [6] J. Holanda *et al.*, "Spin Seebeck effect in the antiferromagnet nickel oxide at room temperature," *Appl. Phys. Lett.*, vol. 111, no. 17, p. 172405, Oct. 2017, doi: 10.1063/1.5001694.
- [7] G. R. Hoogeboom and B. J. van Wees, "Nonlocal spin Seebeck effect in the bulk easy-plane antiferromagnet NiO," *Phys. Rev. B*, vol. 102, no. 21, p. 214415, Dec. 2020, doi: 10.1103/PhysRevB.102.214415.
- [8] R. Cheng, D. Xiao, and A. Brataas, "Terahertz Antiferromagnetic Spin Hall Nano-Oscillator," *Phys. Rev. Lett.*, vol. 116, no. 20, p. 207603, May 2016, doi: 10.1103/PhysRevLett.116.207603.
- [9] P. C. M. Planken, H.-K. Nienhuys, H. J. Bakker, and T. Wenckebach, "Measurement and calculation of the orientation dependence of terahertz pulse detection in ZnTe," *J. Opt. Soc. Am. B*, vol. 18, no. 3, p. 313, Mar. 2001, doi: 10.1364/JOSAB.18.000313.
- [10] H. Qiu *et al.*, "Ultrafast spin current generated from an antiferromagnet," *Nat. Phys.*, vol. 17, no. 3, pp. 388–394, Mar. 2021, doi: 10.1038/s41567-020-01061-7.
- [11] C. Tzschaschel *et al.*, "Ultrafast optical excitation of coherent magnons in antiferromagnetic NiO," *Phys. Rev. B*, vol. 95, no. 17, p. 174407, May 2017, doi: 10.1103/PhysRevB.95.174407.
- [12] F. N. Kholid *et al.*, "The importance of the interface for picosecond spin pumping in antiferromagnet-heavy metal heterostructures," *arXiv preprint*, vol. arXiv:2208.08332, p. 21, 2022, doi: <https://doi.org/10.48550/arXiv.2208.08332>.
- [13] T. Higuchi, N. Kanda, H. Tamaru, and M. Kuwata-Gonokami, "Selection Rules for Light-Induced Magnetization of a Crystal with Threefold Symmetry: The Case of Antiferromagnetic NiO," *Physical Review Letters*, p. 4, 2011.

Reviewers' Comments:

Reviewer #1:

Remarks to the Author:

I am satisfied with the authors' responses. I particularly applaud the efforts that went into growing new samples with Ta and W to demonstrate that the THz emission is due to spin currents. About the non-linear Seebeck effect; the reference Seifert et al. [34] does indeed explain the time-correlated "rectified" effects that can take place, and lead to spin currents, which I hadn't properly understood earlier. I still recommending adding a phrase or two in the present manuscript so that the difference is more clear to the general reader. I therefore recommend publication in Nature Communications.

Jon Gorchon

Reviewer #3:

Remarks to the Author:

I appreciate that the authors have substantially re-written the article and I thank them for all explanations which they provide in their rebuttal letter. The focus of the article is now shifted to the generation of coherent THz peak and its explanation by excitation of the strain.

Personally, I am not completely convinced that thermally-induced phonon excitation with a high intensity laser pulse would produce a coherent response in lattice and then in spins. However, the article opens questions and would probably bring more research in future. I think that the article could be published.

Dear Editor,

We are very pleased from the referee recommendations to publish our work in Nature Communications and to address below the final recommendation of the referees about our manuscript.

Reviewer #1 (Remarks to the Author):

I am satisfied with the authors' responses. I particularly applaud the efforts that went into growing new samples with Ta and W to demonstrate that the THz emission is due to spin currents. About the non-linear Seebeck effect; the reference Seifert et al. [34] does indeed explain the time-correlated "rectified" effects that can take place, and lead to spin currents, which I hadn't properly understood earlier. I still recommending adding a phrase or two in the present manuscript so that the difference is more clear to the general reader. I therefore recommend publication in Nature Communications.
Jon Gorchon

We thank the referee for his comments about our final manuscript and we agreed to add some sentences for a clearer explanation to the reader, we have added the following sentences highlighted in the manuscript:

"Lastly, it should be noted that the symmetry of the (001) films allows for a non-linear SSE [34], proportional to the temperature gradient at the NiO/Pt interface, and arising from time correlated spin-fluctuations in the heavy metal. This contribution (reported in the ferromagnetic insulator YIG capped with Pt [34]) has the same temporal signature as the lattice heating contribution in the Pt, with a spike signal just after the pump pulse followed by a decay in time (see SM6)."

Reviewer #3 (Remarks to the Author):

I appreciate that the authors have substantially re-written the article and I thank them for all explanations which they provide in their rebuttal letter. The focus of the article is now shifted to the generation of coherent THz peak and its explanation by excitation of the strain. Personally, I am not completely convinced that thermally-induced phonon excitation with a high intensity laser pulse would produce a coherent response in lattice and then in spins. However, the article opens questions and would probably bring more research in future. I think that the article could be published.

We thank the referee for reconsidering his/her opinion about our study and for his/her comments about the efforts we have put in rewriting according to his/her suggestions.